# The In Vivo Toxicity Assessments of Water-Dispersed Fluorescent Silicon Nanoparticles in *Caenorhabditis elegans*

**DOI:** 10.3390/ijerph19074101

**Published:** 2022-03-30

**Authors:** Qin Wang, Yi Zhu, Bin Song, Rong Fu, Yanfeng Zhou

**Affiliations:** 1Jiangsu Key Laboratory of Infection and Immunity, Institutes of Biology and Medical Sciences (IBMS), Soochow University, Suzhou 215123, China; qinwang@suda.edu.cn (Q.W.); zhuyi216@suda.edu.cn (Y.Z.); fr131@suda.edu.cn (R.F.); 2Institute of Functional Nano & Soft Materials (FUNSOM) and Collaborative Innovation Center of Suzhou Nano Science and Technology, Jiangsu Key Laboratory for Carbon-Based Functional Materials & Devices, Soochow University, Suzhou 215123, China; bsong@suda.edu.cn; 3School of Public Health, Shanghai Jiao Tong University School of Medicine, Shanghai 200025, China

**Keywords:** fluorescent silicon nanoparticles, *C. elegans*, toxicity, in vivo, endoplasmic reticulum stress, mitochondria stress, oxidative stress, endocytic sorting, host defense

## Abstract

Fluorescent silicon nanoparticles (SiNPs), resembling a typical zero-dimensional silicon nanomaterial, have shown great potential in a wide range of biological and biomedical applications. However, information regarding the toxicity of this material in live organisms is still very scarce. In this study, we utilized *Caenorhabditis elegans* (*C. elegans*), a simple but biologically and anatomically well-described model, as a platform to systematically investigate the in vivo toxicity of SiNPs in live organisms at the whole-animal, cellular, subcellular, and molecular levels. We calculated the effect of SiNPs on *C. elegans* body length (N ≥ 75), lifespan (N ≥ 30), reproductive capacity (N ≥ 10), endocytic sorting (N ≥ 20), endoplasmic reticulum (ER) stress (N ≥ 20), mitochondrial stress (N ≥ 20), oxidative stress (N ≥ 20), immune response (N ≥ 20), apoptosis (N ≥ 200), hypoxia response (N ≥ 200), metal detoxification (N ≥ 200), and aging (N ≥ 200). The studies showed that SiNPs had no significant effect on development, lifespan, or reproductive ability (*p* > 0.05), even when the worms were treated with a high concentration (e.g., 50 mg/mL) of SiNPs at all growth and development stages. Subcellular analysis of the SiNP-treated worms revealed that the intracellular processes of the *C. elegans* intestine were not disturbed by the presence of SiNPs (*p* > 0.05). Toxicity analyses at the molecular level also demonstrated that the SiNPs did not induce harmful or defensive cellular events, such as ER stress, mitochondria stress, or oxidative stress (*p* > 0.05). Together, these findings confirmed that the SiNPs are low in toxicity and biocompatible, supporting the suggestion that the material is an ideal fluorescent nanoprobe for wide-ranging biological and biomedical applications.

## 1. Introduction

With the rapid development of nanotechnology, numerous nanomaterials have been developed for various applications. Recent progress in the application of fluorescent nanomaterials in the field of biomedical sciences has complemented nanobiotechnology with advanced imaging techniques. As an example, fluorescent nanomaterials, such as up-conversion nanophosphors [1,2,3], quantum dots (QDs) [4,5,6,7], carbon [8,9], and silica [10,11], have been exploited as fluorescent probes in biological and biomedical studies. The consensus is that a negligible toxicity of nanomaterials is the prerequisite for extensive biomedical applications which are still poorly satisfied [12,13,14]. Most types of fluorescent nanomaterials display various degrees of cytotoxicity, both in vitro and in vivo [15,16,17,18,19,20], and trigger multiple types of cellular dysfunction, such as endoplasmic reticulum stress, mitochondria stress, and oxidative stress [21,22,23,24].

During the past few years, taking advantage of the low toxicity of silicon, fluorescent SiNPs have been extensively explored as prime candidates for biocompatible fluorescent nanostructures, holding tremendous promise for biosensing and bioimaging applications. Specifically, fluorescent SiNPs have been shown to be superbly suitable for use in long-term and real-time imaging, cellular targeting and trafficking, fluorescence ink, the detection and photodynamic treatment of bacterial infections, and cancer cell detection and therapy [25,26,27,28,29,30]. There have been advances in the understanding of the toxicity of SiNPs in both cell cultures and mammalian animal models, but these knowledges are still fragmented and sometimes inconsistent. For example, several studies have demonstrated that SiNPs present a favorable biocompatibility and low toxicity in vitro [31,32]. Additionally, in vivo toxicity studies have shown that no obvious adverse effects are found in mice or monkeys in terms of weight, blood chemistry, or behavior, even at a large dose of 200 mg/kg [33]. However, further histopathology analyses have revealed the adverse effects of SiNPs in the livers of mice, but have shown no such effects in those of monkeys. It is worth pointing out that before employing SiNPs in practical biomedical applications, further information on these nanostructures’ in vivo toxicity must be collected and analyzed systematically using simple but reliable and efficient model systems.

*C. elegans* is a frequently used model organism in various biological and biomedical studies, due to its well-defined anatomical features and highly conserved, well-characterized genetic background [34,35,36]. In recent years, *C. elegans* has grown in popularity among researchers investigating the toxicity of various functional nanostructures, such as carbon-based nanomaterials [37,38,39,40,41], metal-based nanomaterials [42,43,44,45], silica nanomaterials [46,47], II–VI semiconductor QDs [48,49,50,51], and up-conversion nanocrystals [52,53]. For example, *C. elegans* was utilized as a model organism in order to fully assess the in vivo physiological behavior and toxicity of QDs. The partial degradation of QDs in the digestive system and the transportation of QDs from the digestive system to the reproductive system have been detected, leading to severe reproductive toxicity in *C. elegans* [49]. Chen et al. showed that after exposing nematodes to amide-modified single-walled carbon nanotubes, acute and chronic toxicity were found [39]. Serious reproductive senescence and/or early death due to silver, gold, or silica nanoparticles in *C. elegans* have also been revealed in recent studies [43,44,45,46,47]. On the other hand, graphite nanoplatelets, up-conversion nanocrystals, and fluorescent nanodiamonds have shown no obvious toxic effects in *C. elegans* from growth to procreation [38,52,53,54]. In addition to revealing the toxic effects of nanomaterials on reproduction and development at the whole-animal or single-cell level, *C. elegans* can be used to demonstrate the risks of nanomaterials from the perspective of molecular biology [39,41,44,48].

The aim of this study was to fully evaluate the in vivo toxicity of SiNPs at three different levels: the whole-animal level, the subcellular level, and the molecular level. We hypothesized that fluorescent SiNPs can be promising candidates for use as low-toxicity fluorescent bioprobes, which may provide reliable information during the tracking of physiological or pathophysiological processes in living organisms.

## 2. Materials and Methods

### 2.1. Synthesis and Characterization of Fluorescent SiNPs

Fluorescent SiNPs were synthesized according to the previously reported photochemical strategy [31]. In detail, the precursor solutions were obtained by adding (3-Aminopropyl) trimethoxysilane (APS) to double-distilled water dispersed with 1,8-naphthalimide. The SiNPs were obtained after 1 h UV irradiation. To exclude the residual reagents (APS and 1,8-naphthalimide), the two following steps were performed. Residual 1,8-naphthalimide was first separated from the as-prepared SiNP solution by centrifugation (7000 rpm for 15 min). Subsequently, residual APS in the obtained SiNPs solution was eliminated by dialysis (MWCO: 1000, Spectra/Pro). Purified SiNPs were concentrated by a rotary evaporator. The UV–vis absorption, photoluminescence (PL), transmission electronic microscopy (TEM), and dynamic light scattering (DLS) techniques were used to characterize the as-prepared SiNPs.

### 2.2. C. elegans Strains and Administration

Wild-type strain (N2) and transgenic strains carrying zcIs13[hsp-6::GFP], frls7[Pnlp-29::GFP+Pcol-12::DsRed], zcIs4[hsp-4::GFP], and ldIs3[Pgcs-1::GFP+pRF4(rol-6(su1006))] were purchased from the Caenorhabditis Genetics Center (CGC); qxEx1867[Pvha-6::mCherry::RAB-11] and qxIx110[Pges-1::mCherry::RAB-5] were obtained from Dr. Xiaochen Wang; pwIs112[Pvha-6::hTAC::GFP] and pwIs717[Pvha6::hTfR(short)::GFP] were obtained from Dr. Barth Grant (Rutgers University, Piscataway, NJ, USA). *C. elegans* strains were cultured on standard nematode growth media (NGM) agar plates seeded with OP50 strain bacteria at 20 °C [55]. In this project, synchronized worms were used for all studies. The mixture of sodium hydroxide with sodium hypochlorite was used to lyse gravid adults; then, the age-synchronous worms of the L1 larval stage were obtained. The desired stages of *C. elegans* were obtained by transferring synchronized L1-stage worms to the OP50-seeded NGM plates and feeding worms with OP50 bacteria.

For the induction of oxidative stress, well-fed young-adult-stage worms were exposed to 20 mM hydrogen peroxide (H_2_O_2_) for 4 h followed by 1 h recovery on plates without H_2_O_2_ before samples were collected for imaging [56,57]. For the induction of ER stress, an OP50 suspension containing 5 μg/mL tunicamycin (Solarbio, Beijing, China) was seeded onto NGM plates [58]. Synchronized L2-stage worms were transferred onto tunicamycin-containing NGM plates for 48 h before being collected for imaging. For the induction of mitochondrial stress, L4 worms were transferred onto ethidium-bromide-containing NGM plates (Bio-Rad, Hercules, CA, USA, 25 μg/mL) for 24 h, and then collected for imaging [59]. For the induction of epidermal injury, young-adult-stage worms were cultured on the NGM plates pre-coated with fine glass shards with an average diameter of 50 μm. Worms were allowed to crawl for 4 h and then recover for 1 h before sample collection.

### 2.3. Nanoparticle Uptake Assays

Synchronized worms at the desired stage were transferred onto NGM agar plates seeded with OP50 strain bacteria. Subsequently, 100 μL of SiNP solution (50 mg/mL) or 100 μL of CdTe QD solution (10 μmol/L) was added to each plate. Double-distilled water (H_2_O) was used in place of the nanoparticle solution as a negative control. Worms were cultured on the pre-prepared NGM agar plates for the indicated time span. For short-term treatment, L4-stage larvae were administrated with nanoparticles for 4 h or 24 h before analysis. For long-term treatment, L1-stage larvae were administrated with nanoparticles throughout the entire larval stage until the worms of the negative control group reached the young adult stage.

### 2.4. RT-PCR Analysis

Total cellular RNA was isolated using the RNAiso plus reagent, and cDNA was reverse-transcribed with primescript RT master mix (TakaRa, Shiga, Japan). qRT-PCR reactions were carried out using a Faststart universal SYBR Green master (Roche, Basel, Switzerland) on Eppendorf Mastercycler EP realplex. The list of primers is provided in Table 1. The amplification program for all genes was as follows: pre-incubation at 95 °C for 10 min, followed by 40 temperature cycles for 15 s at 95 °C, 30 s at 60 °C, and 12 s at 72 °C. Each reaction was run in quadruplicates. The original Ct data were exported into REST 2009 software, and the relative gene-expression levels were determined after normalization against the reference gene act-1.

### 2.5. Statistical Analysis

Statistical significance was assessed with the Kruskal–Wallis test (Origin 8.1, Origin Lab Co., Northampton, MA, USA). *p*-values < 0.05 were considered as significant.

## 3. Results

### 3.1. Characterization of SiNPs

The synthesized SiNPs were first examined by their UV–vis absorption spectrum, as displayed in Figure 1a, indicating that the SiNPs had a resolved absorption peak. The normalized PL spectra of the SiNPs are displayed as well (Figure 1c). The inset in the top-right corner of Figure 1c shows the images of SiNP solution under natural light (left) and 405 nm UV laser irradiation (right). The distinct green luminescence signal of the solution under 405 nm laser irradiation further confirmed the strong fluorescence of the synthesized SiNPs. The DLS measurement showed that the hydrodynamic diameter of the as-prepared SiNPs was approximately 5.0 nm (Figure 1b). Notably, particles with a size range of less than 10 nm exhibited rapid renal clearance and then presented low toxicity in vivo, offering great advantages for bioimaging in living organisms [60,61]. The high-resolution transmission electron microscopy (HRTEM) images of SiNPs showed spherical particles with good monodispersity, and the average diameter was~3.7 ± 0.5 nm (Figure 1d). Notably, the particle sizes measured by TEM were smaller than the DLS measurements due to the different surface states of nanoparticles [62]. Specifically, the water in the SiNP samples was clearly removed before the TEM measurements, whereas the DLS characterizations for SiNP samples were directly performed in an aqueous condition. In addition, the very low amounts of organic molecules linked to the surface of the particles could be barely observed by TEM due to their extremely low contrast, but they could be readily detected by DLS, resulting in a difference in the particle size examined by TEM and DLS.

### 3.2. Overall Effects of SiNPs on C. elegans Growth, Longevity, and Reproductivity

Body length, lifespan, and reproduction are classical toxicity indicators in *C. elegans*-related studies [63,64]. To investigate the long-term effect of SiNPs on the development of worms, L1-stage worms were administrated with SiNPs throughout the entire larval stage. Worms treated with H_2_O and CdTe QD served as negative and positive controls, respectively. The bright-field images presented in Figure 2a,b exhibited no obvious difference in morphology between the worms treated with SiNPs and H_2_O. In contrast, the CdTe QD-treated worms displayed severe growth inhibition, which was in agreement with previous reports (Figure 2c) [49,51]. The quantitative analysis of body length also showed that compared with the H_2_O-treated worms, there were no significant changes in the body length of worms, even after they were treated with SiNPs for the whole larval stage. However, a significant decrease in the body length was found in the CdTe QD-treated *C. elegans*, only reaching approximately 40% of the normal value (Figure 2d). These results suggested that the SiNPs did not affect the overall development process of *C. elegans* at the systemic level.

To further explore the biocompatibility of SiNPs, lifespan, and brood size, studies on *C. elegans* exposed to SiNPs were then performed. As shown in Figure 3a, short-term exposure (4 h) to high amounts of SiNPs did not shorten lifespan. Figure 3b demonstrates that the worms exposed to SiNPs for 4 h also showed no deviation from the worms of the negative control group for progeny production, indicating no acute toxicity of SiNPs in vivo. To examine the long-term effects of SiNPs on longevity and reproduction, worms were fed with 50 mg/mL SiNPs for the entire larval stage. Similarly, we did not observe obvious changes in lifespan and progeny production after SiNP treatment as compared with the negative controls (Figure 3c,d). The above data indicate that either the short-term or continued presence of SiNPs in the worm tissues did not interfere with the organism, indicating the low toxicity and good biocompatibility of SiNPs in the living organism.

### 3.3. Subcellular Effects of SiNPs on C. elegans

Although whole-animal analyses revealed no detectable toxic effects of SiNPs on *C. elegans* from the macroscopic view, subcellular analysis was still necessary for the comprehensive investigation of the biocompatibility of SiNPs. When worms were fed with SiNPs, the nanoparticles were mainly taken up by the intestinal cells of worms through the intestinal lumen; hence, we mainly focused on the intestinal cellular processes. Previous study reported that fluorescent CdTe nanoparticles specifically affected early and recycling endocytosis, as well as endocytic cargo transportation, in the *C. elegans* intestine [48]. Therefore, we tested the influence of fluorescent SiNPs on the same biological processes. In healthy worms, either mCherry::RAB-5 fusion-protein-marked early endosomes or mCherry::RAB-11 fusion-protein-labeled recycling endosomes were distributed evenly in punctuated patterns in the intestinal cells (Figure 4a,d) [65]. As shown in Figure 4c,f, consistent with previous reports, large endosome aggregates were found at the apical side in intestinal cells of the CdTe QD-treated worms [48,51]. In contrast, no obvious mCherry::RAB-11 or mCherry::RAB-5 aggregates were found when worms were exposed to SiNPs (Figure 4b,e). To further investigate the potential effect of SiNPs on the endocytosis process in the *C. elegans* intestine, human transferrin receptor (hTfR) and α-chain of the human IL-2 receptor TAC (hTAC) were introduced to monitor protein traffic in the intestinal cells of SiNP-treated worms. These two recycling cargo proteins are markers of clathrin-dependent endocytosis and clathrin-independent endocytosis, respectively [66]. In accordance with previous studies, we found that the hTAC::GFP and hTfR::GFP fusion proteins presented uniform distributions in the intestinal cells of normal *C. elegans* and enrichment at the basolateral membranes (Figure 4g,j) [51]. After exposure to SiNPs for 24 h, hTAC::GFP and hTfR::GFP still exhibited uniform distributions and enrichment at the basolateral membranes (Figure 4h,k), the same as in the control group. In contrast, after treatment with QDs for 24 h, both hTAC::GFP and hTfR::GFP lost basolateral enrichment and formed large aggregates at the apical side of the intestinal cells (Figure 4i,l). Taken together, these results indicate that, unlike fluorescent CdTe QDs, the SiNPs do not disturb the endocytosis processes in *C. elegans* intestinal cells.

### 3.4. Potential of SiNPs to Trigger Self-Protective Pathways in C. elegans

To fully investigate the potential toxic effect of SiNPs at the molecular level, we examined the induction of several major stress responses and defensive pathways of adult *C. elegans*. ER stress, mitochondrial stress, and oxidative stress are the most frequently reported stress responses caused by nanomaterials [67,68,69]. The reporter genes commonly used for monitoring the induction of ER stress, mitochondria stress, and oxidative stress in *C. elegans* are *hsp-4*, *hsp-6*, and *gcs-1*, respectively. *hsp-4* and *hsp-6* encode the *C. elegans* ER and mitochondrial-localized chaperone proteins, respectively [58,70]. *gcs-1* is a phase II detoxification enzyme gene whose transcription is activated in response to oxidative stress in *C. elegans* [71]. As expected, the *hsp-4::GFP* (Figure 5c), *hsp-6::GFP* (Figure 5f), and *Pgcs-1::gfp* (Figure 5i) signals all significantly increased under ER stress, mitochondria stress, and oxidative stress triggered by tunicamycin, ethidium bromide, and H_2_O_2_, respectively. However, even after exposure to SiNPs for 24 h, the levels of *hsp-4::GFP* (Figure 5b), *hsp-6::GFP* (Figure 5e), and *Pgcs-1::gfp* (Figure 5h) fusion proteins were mostly unaltered compared with those of H_2_O-treated worms.

Previous reports have demonstrated that the structural injury of the epidermal barrier of the *C. elegans* could stimulate an immune response and activate the expression of the antimicrobial peptide *nlp-29* (Figure 5l) [72]. According to previous findings, SiNPs can be incorporated into the epidermal layer of *C. elegans* [73]. To investigate whether the SiNPs incorporated in the epidermal layer can cause injury to the epidermis of *C. elegans*, we checked the expression level of *nlp-29* in the SiNP-treated worms. As shown in Figure 5j,k, the treatment of SiNPs did not induce significant changes in *nlp-29* expression in *C. elegans*, implying that the internalized SiNPs did not cause damage to the epidermis of *C. elegans*. These observations were further confirmed by quantitative analysis (Figure 5m), which showed that the incorporation of SiNPs into the *C. elegans* tissues did not trigger major stress responses or host defense processes on a molecular level.

We also examined changes in the expression of several other stress-related genes, representing different pathological conditions after SiNP treatment. APE-1/iASPP is an evolutionarily conserved inhibitor of p53. Its expression can be upregulated by UV radiation or apoptosis [74]. HIF-1 is orthologous to the mammalian hypoxia-induced factor and is activated upon ion toxicity or under low-oxygen conditions [75]. MTL-2 has functions in metal detoxification and stress adaptation, and is upregulated when worms are exposed to heavy metals or heat shock [76]. SGK-1 is an ortholog of mammalian serum and glucocorticoid-inducible kinases (SGKs). The expression levels of *sgk-1* are decreased in aging animals [77]. The RT-PCR analyses revealed that none of the tested genes showed significant changes in expression after SiNP incorporation into *C. elegans* (Figure 6).

## 4. Discussion

With the rapid progress made in the biological applications of functional nanomaterials, the issues of safety and risk for these nanomaterials have aroused considerable interest in recent years, resulting in a large number of studies related to nanobiosafety. Fluorescent SiNPs are regarded as promising bioprobes with a wide range of bioimaging applications due to their outstanding optical properties and the low toxicity of silicon. Recently, a new and powerful photochemical method, capable of the low-cost and high-quantity production of fluorescent SiNPs, was reported by Zhong et al. [31]. Although toxicity studies focusing on in vitro effects have shown a low degree of toxicity for these nanomaterials, additional data on in vivo toxicity must be investigated systematically. *C. elegans* has been widely used as an alternative model organism in order to investigate the biosafety and toxicity of many kinds of functional nanomaterials. Herein, we employed *C. elegans* as an in vivo model to fully evaluate the in vivo toxicity of SiNPs at three different levels. These findings showed that, unlike most other nanostructures examined in *C. elegans* (e.g., carbon nanotubes, silica nanoparticles (NPs), silver NPs, gold NPs, and QDs), the SiNPs induced neither detectable toxicity nor any other apparent side effects in the animals. It is important to point out that, in addition to the pure fluorescent SiNPs, iron-doped SiNPs have also been demonstrated to be promising multifunctional probes for fluorescent and magnetic imaging [78]. This study has also provided a new reference for the wider bioimaging application of iron-doped SiNPs.

## 5. Conclusions

Reliably and comprehensively understanding the biocompatibility of nanostructures in living organisms is one of the critical requirements of nanomedicine. Herein, we chose the feeding method to introduce large quantities of SiNPs into *C. elegans* in order to fully investigate the in vivo toxicity of SiNPs at the organismal, subcellular, and molecular levels. Multiple toxic endpoints (e.g., growth, longevity, reproductivity, endocytosis process, and stress-response gene expression) were analyzed in SiNP-treated *C. elegans*. The results demonstrated that internalized SiNPs did not affect development, viability, or reproductive capacity, nor did they influence physiological activities such as endocytosis. The assays conducted using different stress-response genes showed that the SiNPs did not elicit detectable stress responses in the organisms. The low toxicity of fluorescent SiNPs shown in *C. elegans* presents SiNPs as ideal bioprobe candidates for the long-term tracking or imaging of biological processes in living organisms.

## Figures and Tables

**Figure 1 ijerph-19-04101-f001:**
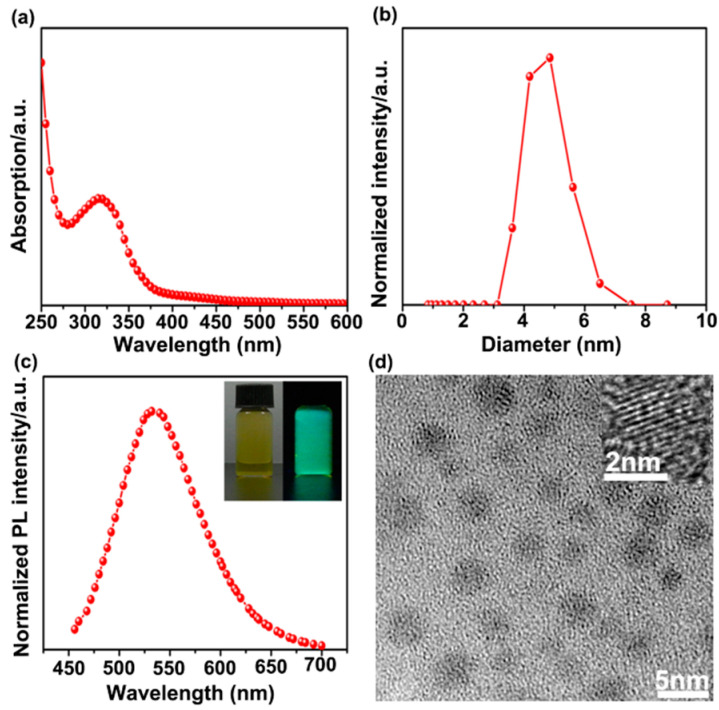
Physicochemical characterizations of SiNPs used in this study. (**a**–**c**) The representative UV absorption (**a**), dynamic light scattering (DLS) histogram (**b**), and photoluminescence (PL) spectra (**c**) of the SiNPs. The inset of (**c**) shows the appearance of SiNPs in water under ambient light (left) and 405 nm laser excitation (right). (**d**) TEM images of SiNPs.

**Figure 2 ijerph-19-04101-f002:**
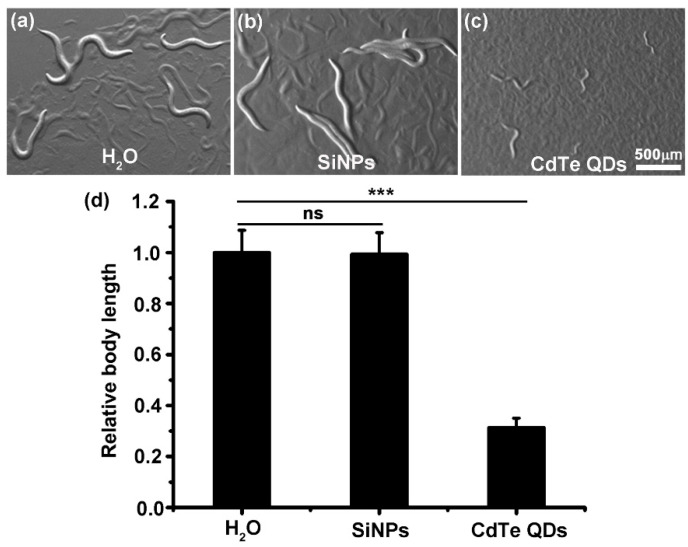
Long-term effects of SiNPs on the development of *C. elegans*. (**a**–**c**) The bright-field images of the *C. elegans* treated with SiNPs, CdTe QDs, or H_2_O, from L1 stage until H_2_O control group reached young adult stage. The same magnification was used in all the images. (**d**) Body length measurements of worms for the different treated groups. N ≥ 75. Error bars represent mean ± SEM; ns, *p* > 0.05, *** *p* < 0.001.

**Figure 3 ijerph-19-04101-f003:**
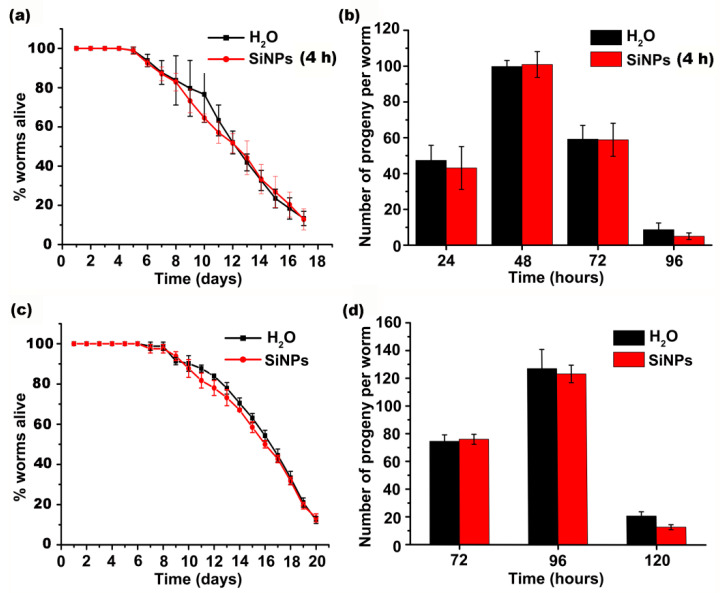
Short-term and long-term effects of SiNPs on the lifespan and reproduction of *C. elegans*. The survival curves (**a**) and the number of laid eggs (**b**) of the *C. elegans* administrated with H_2_O or SiNPs for 4 h from L4 larval stage. The survival curves (**c**) and the number of laid eggs (**d**) of the *C. elegans* continuously administrated with H_2_O or SiNPs from L1 stage until H_2_O control group reached young adult stage. All experiments were repeated independently three times (three biological replicates, N ≥ 30/condition for data in (**a**,**c**), N ≥ 10/condition for data in (**b**,**d**). Error bars represent mean ± SEM. The statistical analysis indicated that there were no significant differences in the lifespan and brood sizes between the H_2_O-treated control group and SiNP-treated group (*p* > 0.05).

**Figure 4 ijerph-19-04101-f004:**
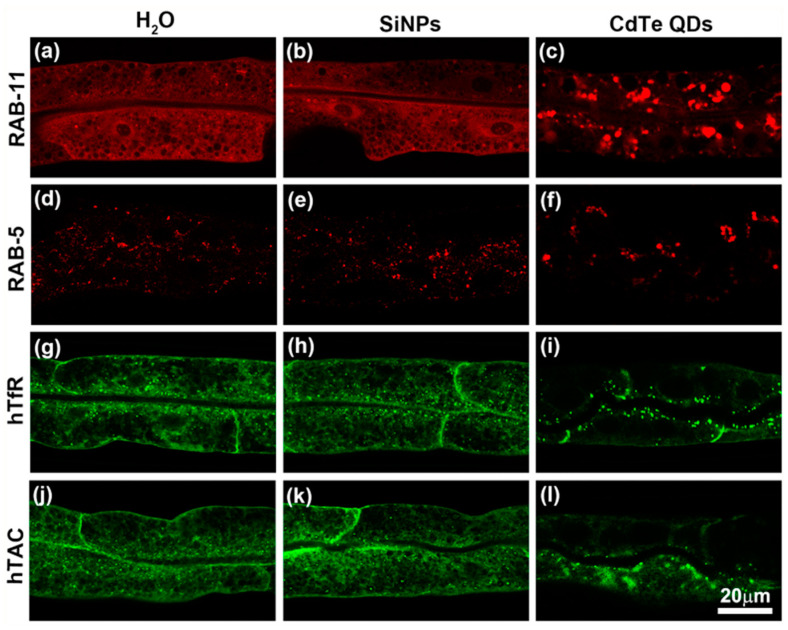
The influence of CdTe QDs and SiNPs on endocytic sorting in *C. elegans* intestinal cells. The confocal images of intestinal endosomes labeled by mCherry::RAB-11 (**a**–**c**) or mCherry::RAB-5 (**d**–**f**) in worms treated with H_2_O (**a**,**d**), SiNPs (**b**,**e**) or CdTe QDs (**c**,**f**) for 24 h. Confocal images of cadherin-dependent cargo hTfR::GFP (**g**–**i**) and cadherin-independent cargo hTAC::GFP (**j**–**l**) distribution within the intestinal cells of H_2_O (**g**,**j**), SiNPs (**h**,**k**) or CdTe QD-treated worms (**i**,**l**). The same magnification was used in all of the images.

**Figure 5 ijerph-19-04101-f005:**
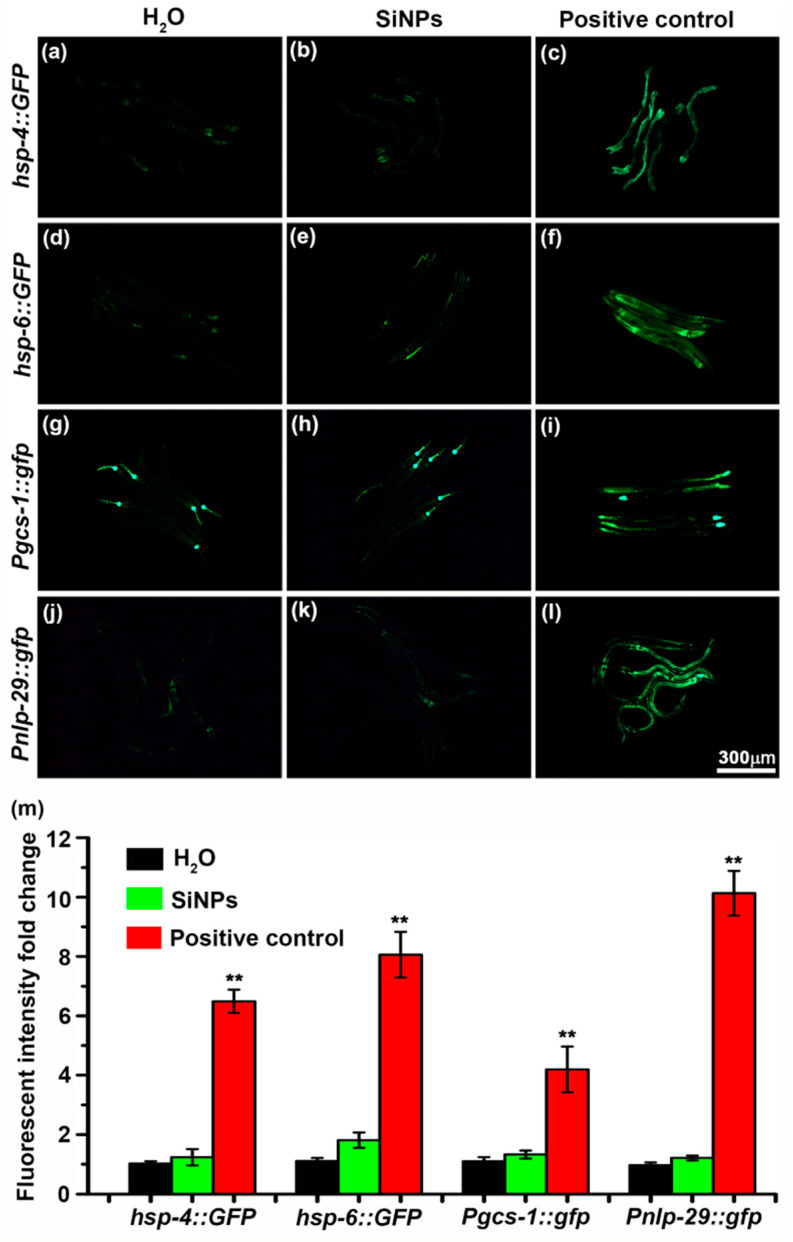
Induction of *C. elegans* major stress or host defense responses by SiNP treatment. Fluorescent images of worms treated with H_2_O (**a**), SiNPs (**b**), and tunicamycin (**c**) with a GFP reporter for ER stress. Fluorescent images of worms treated with H_2_O (**d**), SiNPs (**e**), and ethidium bromide (**f**) with a GFP reporter for mitochondrial stress. Fluorescent images of transgenic worms carrying the GFP reporter for oxidative stress subjected to H_2_O (**g**), SiNPs (**h**), and H_2_O_2_ (**i**) treatment. Fluorescent images of transgenic worms carrying the GFP reporter for innate defense subjected to H_2_O (**j**), SiNP treatment (**k**), and physical injury (**l**). The same magnification was used in all of the images (**m**). Quantitative analysis of fluorescent intensity fold change of worms treated with H_2_O and SiNPs and a positive control group corresponding to (**a**–**l**). N ≥ 20. Error bars represent mean ± SEM; ** *p* < 0.01.

**Figure 6 ijerph-19-04101-f006:**
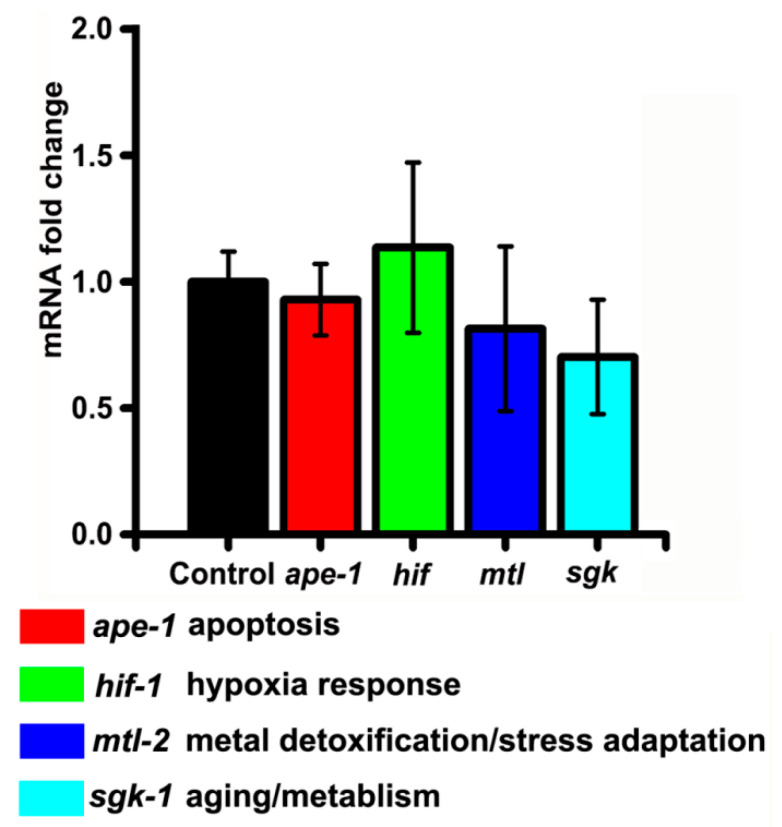
The relative expression levels of other different stress response genes in SiNP-treated *C. elegans* (three biological replicates, N ≥ 200/condition). The black column represents the normalized expression of the target genes in H_2_O-treated group. Error bars represent mean ± SEM. The statistical analysis indicated that no statistically significant differences in gene expression between the SiNP-treated groups and the H_2_O-treated groups were detected (*p* > 0.05).

**Table 1 ijerph-19-04101-t001:** Primers used for qRT-PCR analysis.

Genes	Accession Number	Primer Forward	Primer Reverse
hif-1	180,359	TGTCTTTCCTGGTTCATTCAAA	ATCCGAAACGAAAGTGATGC
mtl-2	179,899	TGCAACACCGGAACTAAAGA	TTAATGAGCAGCCTGAGCAC
sgk-1	181,697	TTCTTCCTTCCGGTTGATTG	TCGTGATCTCGATGAGTGACA
ape-1	179,601	ACCTCGTGCTTCTGTCGAAC	ATGGCTCCGTCGGTATTTTT
act-1	179,535	TCCTTACCGAGCGTGGTTAC	GTTTCCGACGGTGATGACTT

## Data Availability

Data supporting reported results can be obtained from the authors on request.

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
