# Peer review of "The In Vivo Toxicity Assessments of Water-Dispersed Fluorescent Silicon Nanoparticles in Caenorhabditis elegans"

_ijerph, 2022, doi:10.3390/ijerph19074101_

Round 1
Reviewer 1 Report
Despite the importance of the topic of the manuscript. But, several concerns need to be addressed and extensive revisions are required as follows:
1. The abstract lacks quantitative information about the results and research methods, the presentation is confusing, and there are some incorrect statements. There is a paucity of detail given about the experimental protocols and which metrics were calculated.
2. Keywords: more keywords should be added like endoplasmic reticulum stress; mitochondria stress; oxidative stress.
3. Introduction should end with the hypothesis and aim of the work, not the results. Lines 81-92 should be removed and replaced with the hypothesis and aim of the work.
4. Material and methods:
- On what basis the authors have chosen the tested concentration of SiNP solution or CdTe QD solution? Justify with references.
- Why the authors have tested the effect of SiNP after induction of different stressful conditions, not on the normal conditions?
- Table 1 should be completed with either the accession number of analyzed genes or the references for these primers.
- Statistical analysis: many details are missed. How the data were presented as means ± SE or SD. Does data meet the assumption of homogeneity of variances and normal distribution? Clarify if authors run a homogeneity or normality test.
5. Results and discussion are highly recommended to be separated. Also, the results section needs to be refined by deleting all repletion of the methods used like lines 157-159.
6. Line 193: previous reports but one reference only exists.
7. Please rewrite figures legends briefly and do not explain or repeat the results. Also, clarify n=? and data were presented as means ± SE or SD (especially in figure 6).
8. The manuscript needs to be revised for the English and the overall style of writing. The writing style should be formal from the third-person perspective. Do not use us, them, or ours. For instance, the sentence should not begin with an abbreviation like that in line 243 " hTfR ". The whole manuscript needs to be revised by a native English speaker.
9. There is a problem with using abbreviations throughout the manuscript. The full term should be mentioned first with the abbreviation between paresis then the abbreviations should be used throughout the manuscript. E.g. In line 36, quantum dots have been abbreviated as QDs then the full term has been repeated again in line 65. Also, need to abbreviate the endoplasmic reticulum in line 25 in the abstract as it has been repeated again. Such errors have been repeated many times for other abbreviations throughout the manuscript.
Reviewer 2 Report
The manuscript entitled " The in vivo toxicity assessments of water-dispersed fluorescent 2 silicon nanoparticles in Caenorhabditis elegans" by Q. Wang et al. describes the evaluation of Si NPs’ toxicity.
Overall, the paper is well structured and based on the previously established preparation of Si NPs showing luminescence. After addressing a few comments, the paper could be published in IJERPH.
Specific comments that should be addressed:
- As described in this paper and other studies, pure Si NPs are interesting for imaging purposes. The authors should consider adding information of iron-doped silicon nanoparticles in the manuscript offering another dimension for the imaging (ACS Nano 2012, 6, 6, 5596–5604). Since both Si and Fe-based materials have been FDA approved this vital piece of information would benefit the article. At the same time, other Si NP doping would be interesting for similar studies and especially higher doping levels offer more functionalities.
- I am missing a determination of the silica content in the NP material since the methoxysilane is an excellent source for silica. The authors should comment whether there is a surface oxidation or shell formation etc.. This will also support the origin of the PL.
- The authors should clearly state where the photoluminescence originates, since Si NPs in this size regime would most probably show red emission. The authors could refer to a recently published review article (Nanotechnol Rev 2017; 6(6): 601–612).
- The statistics are rather poor in Figure 2 with only 15+ specimen. The authors should improve on this when possible. Moreover, I am missing scale bars in Figure 2a-c.
- Figure 3: scale bars should be added for each image or properly described that the same magnification has been used in all the images.
- The authors should carefully spell check the manuscript since quite a few errors can be easily found (typos and grammar).
Reviewer 3 Report
The authors presented fluorescent silicon nanoparticles and investigated their in vivo toxicity in living organism.
The authors should reflect and correct the following statement:
"Organic molecules on the surface of the particles could barely
be observed by TEM owing to their extremely low contrast, but its could be readily detected by DLS, resulting in a difference particle sizes examined by TEM and DLS."
In several studies, nanoparticles exclusively made from organic molecules are observed with TEM with accuracy.
The discrepancy in size warrants a stability studies on the nanoparticle in different medium since it is designed to be use as a nanoprobe.
Round 2
Reviewer 1 Report
No further comments to be addressed